# Novel Transcript Discovery Expands the Repertoire of Pathologically-Associated, Long Non-Coding RNAs in Vascular Smooth Muscle Cells

**DOI:** 10.3390/ijms22031484

**Published:** 2021-02-02

**Authors:** Matthew Bennett, Igor Ulitsky, Iraide Alloza, Koen Vandenbroeck, Vladislav Miscianinov, Amira Dia Mahmoud, Margaret Ballantyne, Julie Rodor, Andrew H. Baker

**Affiliations:** 1Centre for Cardiovascular Science, Queen’s Medical Research Institute, University of Edinburgh, 47 Little France Crescent, Edinburgh EH16 4TJ, UK; s1795508@ed.ac.uk (M.B.); vlad.miscianinov@biohabit.co.uk (V.M.); amahmoud@exseed.ed.ac.uk (A.D.M.); magz216@hotmail.com (M.B.); julie.rodor@ed.ac.uk (J.R.); 2Department of Biological Regulation, Weizmann Institute of Science, Rehovot 76100, Israel; igor.ulitsky@weizmann.ac.il; 3Inflammation & Biomarkers Group, Biocruces Bizkaia Health Research Institute, Cruces Plaza, 48903 Barakaldo, Spain; iraide.alloza@ehu.eus (I.A.); k.vandenbroeck@ikerbasque.org (K.V.); 4Ikerbasque, Basque Foundation for Science, 3 María Díaz Haroko Kalea, 48013 Bilbao, Spain

**Keywords:** vascular smooth muscle cells, long non-coding RNAs, RNA sequencing, enhancers

## Abstract

Vascular smooth muscle cells (VSMCs) provide vital contractile force within blood vessel walls, yet can also propagate cardiovascular pathologies through proliferative and pro-inflammatory activities. Such phenotypes are driven, in part, by the diverse effects of long non-coding RNAs (lncRNAs) on gene expression. However, lncRNA characterisation in VSMCs in pathological states is hampered by incomplete lncRNA representation in reference annotation. We aimed to improve lncRNA representation in such contexts by assembling non-reference transcripts in RNA sequencing datasets describing VSMCs stimulated in vitro with cytokines, growth factors, or mechanical stress, as well as those isolated from atherosclerotic plaques. All transcripts were then subjected to a rigorous lncRNA prediction pipeline. We substantially improved coverage of lncRNAs responding to pro-mitogenic stimuli, with non-reference lncRNAs contributing 21–32% for each dataset. We also demonstrate non-reference lncRNAs were biased towards enriched expression within VSMCs, and transcription from enhancer sites, suggesting particular relevance to VSMC processes, and the regulation of neighbouring protein-coding genes. Both VSMC-enriched and enhancer-transcribed lncRNAs were large components of lncRNAs responding to pathological stimuli, yet without novel transcript discovery 33–46% of these lncRNAs would remain hidden. Our comprehensive VSMC lncRNA repertoire allows proper prioritisation of candidates for characterisation and exemplifies a strategy to broaden our knowledge of lncRNA across a range of disease states.

## 1. Introduction

The principal role of vascular smooth muscle cells (VSMCs) in their differentiated state is to provide contractile force in the vessel wall to ensure proper circulation. However, a high level of plasticity relative to other cell types is well established [1,2,3], with phenotypes, such as proliferation, migration, and extracellular matrix production, often displayed at the expense of contractility. This adaptability can aid vessel growth and repair in response to a wide range of biochemical signals or mechanical stresses [1]. Conversely, it also contributes to vessel wall remodelling during some of the most prevalent and life-threatening cardiovascular diseases, such as atherosclerosis, pulmonary hypertension, and restenosis [2]. The molecular mechanisms controlling the wide variety of phenotypic changes involved in such diseases are controlled by both coding and non-coding genes at both the genetic and epigenetic level [3,4]. However, they are not yet understood sufficiently to effectively target them therapeutically.

Long non-coding RNAs (lncRNAs), RNA transcripts > 200 bp in length that do not produce proteins, are mostly uncharacterised. Yet several have been found to be key to a range of vital cellular processes via contributing to epigenetic, transcriptional, or translational regulation [5,6]. However, only a few hundred are experimentally characterised out of tens of thousands annotated through genome-wide sequencing efforts [7]. Therefore, lncRNAs represent a potential cache of novel mechanistic processes crucial for cell function. In addition, many lncRNAs show a high level of specificity in terms of their expression across tissues, cells, and timepoints during development or stimuli responses [8,9], raising the prospect of using them as markers or therapeutic targets to tackle aberrant cell behaviour. For example, recent studies have described *SMILR* [10], *MYOSLID* [11], and *SENCR* [12] with biased expression to VSMCs or the vasculature and as key regulators of cell cycle, migration and differentiation state. However, lncRNA discovery in VSMCs remains limited and relatively unexplored. Therefore, the overall scale of lncRNA contribution to aberrant VSMC behaviour during tissue remodelling is still an open question that crucially needs to be addressed.

An obstructing factor in determining lncRNA regulation of VSMCs is that the annotation of such genes is incomplete, even across the extensively annotated human genome [13]. Even reference annotations, such as GENCODE [14], considered gold standard [13], and the more extensive FANTOM CAT [15], which integrates 5 smaller reference annotations, are missing lncRNAs. The datasets used to create such annotations cannot represent all possible biological and pathological settings so reference annotations are inherently incomplete. In addition, low abundance and lack of poly-adenylated tails for some lncRNAs means many are difficult to detect in transcriptomic datasets, which are polyA-enriched or of insufficient sequencing depth. A tendency for cell-specific expression also hinders their detection in samples containing heterogenous mixtures of cell types, such as tissues. Accordingly, several efforts have aimed to expand human lncRNA annotations beyond reference annotations. For example, building transcripts de novo from sequencing data was found to yield a substantially greater number of lncRNAs of interest in contexts, such as erythropoiesis [16], formation of CD8+ memory T cells [17], and psoriatic skin tissue [18]. Novel lncRNAs obtained from such approaches have demonstrated particularly high expression specificity, in particular cell type specificity and/or high likelihood of differential expression between conditions or stimuli. They have also shown a particular association with enhancer sites, activators of localised transcription often involved in epigenetic control of cellular states [19]. Enhancers that produce lncRNAs show increased signs of influence on neighbouring transcription relative to other enhancers. These lncRNAs are likely to aid formation of regulatory complexes at many of these enhancer sites [20]. Efforts to improve human VSMC lncRNA annotation include studies on coronary artery VSMCs (caSMCs) maintained in standard growth conditions [12] or pushed toward a differentiated phenotype via overexpression of *MYOCD* [11], which together highlighted the pro-contractile lncRNAs *SENCR* and *MYOSLID*. However, no novel transcript discovery methods have yet been applied to VSMCs responding to stimuli that induce proliferative or pro-inflammatory phenotypes. This means a subset of lncRNAs with potentially crucial roles in VSMC-directed tissue remodelling could remain hidden.

To obtain a more complete representation of the lncRNAs expressed in pathologically active VSMCs, we obtained three published RNA sequencing (RNAseq) datasets describing in vitro stimulation of VSMCs into proliferative, migratory, or pro-inflammatory phenotypes or isolated from diseased tissue in vivo. We then combined a novel transcript discovery approach with a stringent lncRNA annotation pipeline, thereby markedly improving the coverage of stimuli-responsive, VSMC-enriched and enhancer-transcribed lncRNAs within VSMC pathology. Our study highlights lncRNAs with high potential to control VSMC pathological states.

View our data interactively at: https://bakergroup.shinyapps.io/VSMClncRNAannotation/.

## 2. Results

### 2.1. A Bioinformatic Approach to Provide a More Complete Annotation of lncRNAs Expressed in VSMCs in Basal and Pathological Conditions

Several LncRNAs are known to be involved in VSMC phenotypic transitions occurring in vessel wall remodelling [10,11,12]. However, a full accounting of lncRNAs expressed in these transitions does not yet exist. Accordingly, to gain in-depth representation of the lncRNAs expressed in pathological VSMCs, we applied a transcript discovery pipeline to published VSMC RNAseq datasets selected based on specific criteria (Figure 1a). We focused on high depth, paired-end total RNA sequencing datasets to identify all lncRNAs and their gene structures (including non-polyA tailed and/or lowly expressed lncRNAs) and we selected two in vitro datasets fulfilling these criteria. The first dataset describes primary human saphenous vein SMCs (svSMCs) either quiesced in 0.2% FBS or treated with interleukin-1α (Il-1α) and/or platelet-derived growth factor-BB (PDGF-BB) [10]. The second dataset describes primary aortic (aoSMCs) or coronary artery (caSMCs) VSMCs, plated in 5% FBS media onto soft or stiff culture matrices [21]. These conditions model a convergence of pro-inflammatory and pro-mitogenic signals, or mechanical stretch in the vessel wall, both of which promote proliferation and disruption of contractility. We also selected an in vivo dataset describing VSMCs isolated and sequenced directly from enzymatically digested carotid plaques derived from symptomatic or asymptomatic patients, defined as such based on lumen size and occurrence of cardiovascular events prior to surgery [22]. Together, these three datasets document a broad span of VSMC types and phenotypes contributing to vessel wall remodelling.

The transcriptome analysis used GENCODE annotation as a reference and included a transcript assembly step to further identify transcripts not previously described in GENCODE (newly assembled transcripts). This approach allows the identification of novel isoforms for GENCODE genes, but also allows the identification of novel genes (newly assembled genes). This analysis was carried out independently for the three datasets. These expanded transcriptomes consisted of ~80,000–90,000 transcripts in total with 0.6–1.5% transcribed from newly assembled genes (Appendix A). To identify high confidence lncRNAs from these complete transcriptomes, we used “Pipeline for Annotation of LncRNAs” (PLAR) [23], which filters lowly-expressed or artefactual transcripts and assesses coding potential based on three distinct tools. As expected, the bulk of expressed transcripts were annotated as protein coding (Appendix A). However, a high confidence set of ~2500–3000 lncRNA annotations were predicted within each transcriptome, with 6–7% deriving from newly assembled gene loci (Appendix A). Our analysis of robustly expressed genes showed that newly assembled lncRNA genes produced transcripts with comparable lengths to transcripts from GENCODE protein-coding genes (PCGs) or GENCODE lncRNA genes (Appendix A). Of these genes, newly assembled and GENCODE lncRNAs have a lower expression compared to PCGs, as expected. Further, the GENCODE lncRNA genes were also more abundant than the newly assembled lncRNA genes but only by a median difference of ~1 FPKM (Appendix A).

To show the validity of the transcript discovery pipeline for lncRNA identification, we assessed if the newly assembled lncRNA transcripts identified in the three datasets were observed in other reference databases. Using GFFcompare [24], we cross-referenced expressed GENCODE and newly assembled lncRNA transcript structures to transcripts annotated in FANTOM CAT [15], a particularly extensive reference annotation. We observed 72% of GENCODE and 40% of newly assembled lncRNAs matched to a FANTOM transcript containing the exact same chain of introns whilst another 25% of GENCODE and 40% of newly assembled lncRNAs contained at least 1 matching splice junction site (Figure 1b). The validation of the complete or partial gene structures for a large proportion of the newly assembled lncRNAs in other annotation sets (derived from other contexts) provide confidence in the identified transcripts and evidence of the lncRNA expression in different datasets. To find further corroborating evidence of transcription for our lncRNAs, we used FANTOM CAT CAGEseq data which accurately defines transcription start sites (TSSs) in ~1800 distinct human samples through sequencing the site of RNA 5′ capping [15]. We identified 74.3% of newly assembled and 86.9% of GENCODE lncRNA genes across all datasets matched to experimentally validated TSSs in FANTOM CAGEseq data (hereafter referred as CAGE-matched lncRNAs) (Figure 1c). The position of these CAGEseq matches indicates the first exons of newly assembled lncRNAs from our analysed datasets were largely complete at their 5′ ends (incomplete by median of 8% of their initial size) (Appendix A).

To gain perspective on the in vivo relevance of newly assembled lncRNAs, we assayed their expression in an RNAseq dataset of carotid plaque tissue. VSMCs are major components of atherosclerosis plaque, with many demonstrating phenotypic modulation [25]. To assess all newly assembled lncRNAs simultaneously, we merged the three expanded annotations into a non-redundant transcriptome containing 255 newly assembled lncRNA transcripts from 207 lncRNA genes (Figure 1d). Analysis of the plaque-derived RNAseq with this non-redundant merged transcriptome demonstrated that 50 (24%) of the newly assembled lncRNA genes were detectable in whole plaque tissue. This is a substantial detection rate, considering that plaques are heterogenous and contain non-VSMC cells contributing to the RNAseq. In addition, the newly assembled lncRNAs were identified in VSMCs from different vessel types grown in distinct conditions and so might not be expressed in plaques. Interestingly, these 50 lncRNAs come from all four independent transcriptomes (Figure 1e), showing each new annotation provides in vivo relevant transcripts. Notably, 24 of these lncRNAs were identified exclusively using the svSMC dataset showing this annotation particularly improved coverage of plaque-expressed lncRNAs.

Together, these analyses expand the representation of lncRNAs expressed in basal and pathological VSMCs in vitro and in vivo and provide confidence in the newly assembled gene structures.

### 2.2. Newly Assembled Genes Substantially Increase the Number of VSMC lncRNAs Detected in Response to Pathological Stimuli

With confidence in our expanded lncRNA annotation, we next sought to comprehensively identify all lncRNAs differentially expressed between conditions in the RNAseq datasets using DESeq2 [26]. For the svSMC dataset, we identified 162 differentially expressed lncRNAs between control and IL-1α/PDGF-BB stimulation (absolute fold change > 1.5, *p* < 0.05 *) out of the 598 robustly expressed lncRNA genes. Notably, newly assembled genes represented 32% of differentially expressed lncRNAs, more than would be expected by chance considering their proportion of total expressed lncRNAs (18%, *p* < 0.0001 ****, Fisher’s exact test) (Figure 2a). Differential expression dynamics were validated via qRT-PCR in the same svSMC proliferation model for 7 GENCODE and three newly assembled lncRNAs chosen from the top 10% lncRNAs with the highest fold changes (Appendix A). We obtained a high and significant correlation between the qRT-PCR and RNA-seq fold changes (R = 0.97, *p* < 0.0001 ****) (Figure 2b). In the VSMC response to stiff culturing dataset, we identified 143 out of 551 and 168 out of 539 differentially expressed lncRNAs for aoSMC and caSMC, respectively. Again newly assembled lncRNAs made up a larger proportion of the differentially expressed.

lncRNAs then would be expected by chance, considering their proportion of total expressed lncRNAs (Figure 2c,d) (25% vs. 15%, *p* < 0.001 *** and 21% vs. 15%, *p* < 0.01 **, Fisher’s exact test for aoSMC and caSMC, respectively). This indicates a particular tendency for newly assembled lncRNA genes to respond to IL-1α/PDGF-BB or increased vascular stiffness, both pathologically associated as pro-mitogenic stimuli.

For the plaque-isolated VSMC dataset, though a comparable portion of expressed lncRNAs were newly assembled (16%), only 4 GENCODE lncRNAs and no newly assembled lncRNA genes were found differentially expressed between VSMCs from symptomatic and asymptomatic plaques. This small number of lncRNAs reflects the smaller number of transcriptional changes in between this dataset, even for protein coding genes (PCGs) (<1% of expressed genes) (Appendix A), likely explained by the higher heterogeneity of plaque-derived VSMCs compared to cultured VSMCs.

To gain perspective on the role the differentially expressed lncRNAs may play in determining VSMC phenotypic state, we hierarchically clustered all differentially expressed genes based on their expression profile across all four conditions in either the saphenous vein-based model (Figure 2f) or the ao/caSMC stiff culture-based model (Figure 2g). We identified six clusters of gene expression changes in each dataset and used gene ontology analysis (using goseq [27]) to associate them with biological processes, cellular components and molecular functions (Figure 2h,i) (Appendix A). GENCODE and newly assembled lncRNAs were found in most clusters, suggesting the contribution of distinct lncRNAs across the different identified processes. We noted a large proportion of newly assembled lncRNAs in cluster 3 and 4 of the svSMC dataset, involved in cytokine and immune response, and in cluster 6 of the ao/caSMC dataset, involved in ribosome, Cajal body and mitochondrial activity. This indicates that newly assembled lncRNAs may be particularly relevant to these specific processes or compartments.

Together our differential expression analysis showed that newly assembled lncRNAs were more likely to be stimuli-responsive than reference lncRNAs, highlighting the importance of novel transcript discovery in pathological contexts.

### 2.3. Novel Transcript Discovery Increases the Representation of VSMC-Enriched lncRNAs with Pathological Association

Genes with cell type and/or state specific expression can hold particular functional relevance in the condition to which their expression is biased [28], and could be targeted by gene therapy approaches with minimal effects on neighbouring cells. Cell type-enriched lncRNAs could be less likely to be annotated in GENCODE than ubiquitously expressed lncRNAs. We therefore sought to examine whether by identifying genes absent from GENCODE, we concordantly increase the coverage of VSMC-enriched lncRNAs.

To assess the tendency for cell-type enriched expression of GENCODE and newly assembled lncRNAs, we again used the FANTOM CAT CAGEseq library, which contains expression data for 69 primary cell categories (including 9 VSMC subtypes) and 174 tissue categories consisting of 744 samples in total. The expression of 801 of the CAGE-matched lncRNAs (defined in Figure 1d) was accessible in FANTOM data (Appendix A). The cell-type specificity of newly assembled and GENCODE lncRNAs was assessed by obtaining enrichment values for each lncRNA within each primary cell category compared to all other primary cell categories. We noted higher enrichment values for newly assembled lncRNAs in a select group of mesenchymal cell type categories including VSMCs. Enrichment values were much lower in several other primary cell categories, including leukocytes, endothelial, and epithelial cells (Appendix A). In contrast, GENCODE lncRNAs enrichment was less variable across cell types (Appendix A), suggesting the higher specificity of expression of newly assembled lncRNAs compared to GENCODE lncRNAs, which were more ubiquitous.

We next aimed to identify VSMC-enriched lncRNAs. FANTOM CAT has previously defined genes with cell-type enriched expression as those with a 5-fold or greater enriched expression in a given category when compared to all other categories [15]. Therefore, by selecting CAGE-matched lncRNAs with enriched expression in at least one VSMC type in the FANTOM CAT library, we were able to identify 72 VSMC-enriched lncRNAs across all datasets. Amongst the CAGE-matched lncRNAs, significantly more newly assembled lncRNAs were found to be VSMC-enriched than would be expected by chance considering their proportion of total expressed lncRNAs in svSMC (31% of VSMC-enriched vs. 12% of expressed lncRNAs, *p* < 0.001 ***, Fisher’s exact test), caSMC (31% of VSMC-enriched vs. 10% of expressed lncRNAs, *p* < 0.001 ***, Fisher’s exact test) or plaque SMCs (20% of VSMC-enriched vs. 10% of expressed lncRNAs, *p* = 0.03 *, Fisher’s exact test) (Figure 3a,c,d). This trend was maintained albeit with borderline significance for aoSMCs (20% vs. 10%, *p* = 0.07, Fisher’s exact test) (Figure 3b). Overall, this demonstrates that the newly assembled lncRNAs have a greater tendency for VSMC-enriched expression when compared to GENCODE lncRNAs.

To reveal the contribution of VSMC-enriched lncRNAs to VSMC phenotypic modulation, we evaluated their likelihood of differential expression in response to pathological stimuli. In the svSMC dataset, we found an increased tendency for VSMC-enriched lncRNAs to be responsive to IL-1α/PDGF-BB as compared to other lncRNAs (18% of differentially expressed vs. 8% of expressed lncRNAs, *p* < 0.001 ***, Fisher’s exact test) (Figure 3e). This effect was also observed if considering lncRNAs responsive to stiffness in aoSMC (10% of differentially expressed vs. 6% of expressed lncRNAs, *p* = 0.03, Fisher’s exact test) and caSMC (13% of differentially expressed vs. 8% of expressed lncRNAs, *p* = 0.02, Fisher’s exact test) (Figure 3e–g), but not in the plaque VSMC dataset (as expected due to the low number of differentially expressed genes). VSMC-enriched lncRNAs are therefore particularly likely to be involved in regulating VSMC response to physiological stimuli.

In total, we identified 37 VSMC-enriched lncRNAs, including 17 newly enriched lncRNAs, responding to either IL-1α/PDGF-BB or stiff-culturing (Figure 3i). The differential expression of these lncRNAs appeared mostly exclusive to a specific dataset with only four lncRNAs upregulated by both IL-1α/PDGF-BB in svSMC and stiff culturing in aoSMC or caSMC. This suggests that the VSMC-enriched lncRNAs respond to different stimuli or are expressed and regulated in specific VSMC subtypes. We also noted that 13 of the VSMC-enriched lncRNAs induced with IL-1α/PDGF-BB in svSMCs were mostly absent in quiescent svSMCs and robustly expressed in vivo in plaque-isolated VSMCs. Together this expression pattern strongly implicates the involvement of these lncRNAs in VSMC transitions to pathological states. In support of this, one lncRNA showing this pattern, SMILR, has known VSMC enrichment along with roles in both VSMC proliferation and vessel wall remodelling [10].

Interestingly, of all 37 VSMC-enriched, differentially-expressed lncRNAs, many showed a greater VSMC enrichment value than *SMILR* [10] and *SENCR* [12] the 2 lncRNAs in the list already characterised as functional and VSMC-enriched. This indicates existence of several lncRNAs with particularly high VSMC expression bias. For example, newly assembled lncRNA *VSMClnc6* and GENCODE lncRNAs *NLGN4Y*-*AS1* and *AC002480.4*, all induced by IL-1α/PDGF-BB, show a VSMC enrichment greater than 20-fold. In addition to their high VSMC enriched expression, some lncRNAs were also expressed to a lesser extent in arterial or cardiac fibroblasts, other mesenchymal types or other muscle types (Figure 3j and Appendix A), suggesting related function in these cell types. Many also show expression bias to certain VSMC subtypes suggesting they may have some specialised function in particular vascular beds. Of note, VSMC-enrichment does not preclude lncRNAs from enrichment in other cell types including those involved in vessel wall remodelling. However, though some expression in vascular endothelial cells was observed this was largely confined to seven lncRNAs. Very little expression was seen across epithelial cells, non-vascular SMCs, leukocytes or pericytes.

Taken together, we show that newly assembled lncRNAs have a greater tendency for VSMC-enriched expression. In turn, VSMC-enriched lncRNAs are substantially associated with VSMC response to IL-1α/PDGF-BB or stiff-culturing. These pathological-associated and VSMC-enriched lncRNAs, therefore, represent potential candidate for the therapeutic targeting of VSMC phenotypic changes.

### 2.4. Novel Transcript Discovery in VSMCs Increases Evidence of Enhancer-Transcribed lncRNAs

As non-reference lncRNAs have previously shown a particular association with enhancers [18] and lncRNA-producing enhancers have shown greater signs of activity influencing neighbouring PCG expression [20], we aimed to identify elncRNAs and their potentially regulated PCGs.

To identify elncRNAs, we selected lncRNAs with a 5′ region overlapping a GeneHancer [29] enhancer site or matched to a CAGE site previously classed as an “elncRNA” in FANTOM CAT (Appendix A). We found 110, 90, and 87 expressed elncRNAs in the svSMC, aoSMC or caSMC datasets, respectively. Newly assembled lncRNAs made up a higher proportion of the elncRNAs than would be expected by chance considering their proportion of all expressed lncRNAs (26% elncRNAs vs. 18% expressed lncRNAs *p* = 0.01, 23% elncRNAs vs. 15% expressed lncRNAs *p* = 0.01, 23% elncRNAs vs. 15% expressed lncRNAs *p* = 0.02, for svSMC, aoSMC, and caSMC respectively, Fisher’s exact test) (Figure 4a–c). Hence, the newly assembled lncRNAs were particularly likely to be enhancer-transcribed.

We also observed a higher proportion of elncRNAs were differentially expressed in response to IL-1α/PDGF-BB than would be expected by chance compared to their proportion of all expressed lncRNAs (Figure 4d; 31% differentially expressed lncRNAs vs. 18% expressed lncRNAs *p* < 0.0001 ****, Fisher’s exact test). This was also true for lncRNAs differentially expressed in response to stiff culturing in aoSMC (Figure 4e; 24% vs. 16% *p* < 0.01 **) and caSMC (Figure 4f; 21% vs. 16% *p* = 0.02 **). ElncRNAs are therefore more likely than other lncRNAs to be differentially expressed in VSMCs responding to these pathological stimuli, with IL-1α/PDGF-BB in particular eliciting a large elncRNA response. Around 34% of these regulated elncRNAs were newly assembled (Figure 4g).

ElncRNAs may increase the activity of their associated enhancer thereby promoting expression of a proximal PCG [20]. To identify candidate PCGs that may be regulated in this manner, we identified all PCGs located within 250 kbp of a differentially expressed elncRNA and co-induced or co-repressed with the elncRNAs (Appendix A). We found candidate PCG targets for 55% of IL-1α/PDGF-BB-responsive elncRNAs and 39% of stiffness-responsive elncRNAs in aoSMC or caSMC (Figure 4h and Appendix A). For the svSMCs and aoSMCs, the number of candidate PCG targets were doubled as a result of the inclusion of newly assembled elncRNAs (Figure 4i).

To find external evidence supporting the regulation of these PCGs by elncRNAs, we assessed if any were linked to elncRNAs in GeneHancer or FANTOM CAT interaction data. GeneHancer interaction annotations are based on (1) physical association with a PCG promoter (using Capture Hi-C data); (2) presence of SNPs linked to changes in PCG expression (expression quantitative trait loci—eQTLs); (3) presence of motifs shared with a nearby PCG promoter for use by a co-expressing transcription factor; or (4) production of enhancer RNAs that co-express with nearby PCGs. In FANTOM CAT, lncRNA/PCG pairs are linked by eQTL-associated SNPs [15]. We found 13 differentially expressed elncRNAs (6 newly assembled) with evidence linking them to candidate PCG targets (Appendix A). Interestingly, 6 out of these 13 candidate PCG targets have been previously shown to contribute to SMC pathology or to be involved in pathological proliferative phenotypes (Table 1). For example, elncRNAs *VSMClnc6* and *AC002480*.*3* are both induced by IL-1α and PDGF-BB in svSMC and linked to expression of the cytokine IL-6 and the chemokine CXCL8 via GeneHancer interaction data (Figure 4j,k). IL-6 and CXCL8 are key pro-inflammatory mediators known to be induced by IL-1α and characterised as promoting VSMC proliferation, migration, pro-inflammatory activity, and vascular remodelling [30,31,32,33,34]. As mentioned above, VSMClnc6 has a high VSMC enrichment value, raising the prospect that this elncRNA may be part of a VSMC-enriched mechanism regulating the expression of *CXCL8*.

## 3. Discussion

In this study, we used novel transcript discovery and a rigorous lncRNA prediction pipeline on two in vitro VSMC and one in vivo VSMC RNAseq datasets to expand the lncRNA annotation of VSMCs in pathological states. We identified 61–109 newly assembled lncRNAs expressed in each of these datasets. Interestingly, these newly assembled lncRNAs were more likely to be differentially expressed in response to pathological stimuli, to be VSMC-enriched, and/or transcribed from enhancer regions (elncRNAs), compared to previously annotated lncRNAs. As enhancers regulate local transcription, we also predicted neighbouring PCGs regulated by the elncRNAs and show many of these PCGs are involved in VSMC pathology. Taken together, we demonstrate that inclusion of non-reference transcripts is crucial to get a complete representation of lncRNAs and their contribution to the regulation of VSMC pathology. VSMCs in such states are rarely used to annotate non-reference transcripts and so our expanded repertoire is a unique source of potential regulatory mechanisms that could contribute to many vascular pathologies.

Both in vitro datasets analysed in this study model increased VSMC proliferation [10,21] and this was confirmed in our analysis through identification of clusters of upregulated genes associated with cell-cycle processes. The presence of lncRNAs in these clusters shows their potential role in regulating proliferation. We also predict that two elncRNAs within the IL-1α/PDGF-BB induced cell-cycle cluster, *AC002480*.*3* and *VSMClnc6*, regulate the cytokine *IL-6* and chemokine *CXCL8* genes respectively. These two factors are known to promote VSMC proliferation [31,33], yet are also indicative of the pro-inflammatory activity induced by IL-1α in the svSMC dataset. Activation of pro-inflammatory cascades likely aids the proliferation of VSMCs, in part through a synergistic potentiation of their response to PDGF-BB [39]. Additionally, VSMC pro-inflammatory activity is at the core of vessel wall remodelling contexts where a sustained inflammatory response, often from senescent VSMCs, promotes the influx of myeloid cells [3]. We see clusters of genes associated with cytokine response induced with IL-1α alone or combined IL-1α/PDGF-BB stimulation and a particularly large number of newly assembled lncRNAs were found in these clusters. This suggests we also capture lncRNA activity involved in IL-1α-driven pro-inflammatory VSMC phenotypes, as well as cell-cycle processes.

In addition to the *IL-6*- and *CXCL8*-associated elncRNAs, we were able to identify several other elncRNAs likely to regulate PCGs with established roles in VSMC pathology. The elncRNA *LINC00973* for instance is co-induced during IL-1α/PDGF-BB stimulation with *DCBLD2*, a PCG known to regulate PDGFR surface levels to promote VSMC proliferation [35]. *GLS* and *NR2F2* (aka *COUPTFII*) are two particularly notable elncRNA-associated PCGs that are co-repressed with IL-1α/PDGF-BB and are involved in promoting pro-fibrotic activity in myofibroblasts (*GLS* [37]) or defining mesenchymal lineage in the vasculature (*NR2F2* [38]). ElncRNAs are only one of many types of components involved in changing the chromatin accessibility at sites associated with VSMC pathology [4]. Therefore, any further characterisation of the elncRNAs highlighted here must put their potential mechanism in context with any recruitment of transcription factors or histone-modifying enzymes.

Our approach to fully define lncRNA contribution could be used in transcriptomics analysis of other contexts of VSMC pathological activation that remain to be explored. For instance subsets of VSMCs in the vessel wall appear particularly prone to phenotypic modulation including those derived from adventitial stem cells and responsible for neointimal VSMC proliferation in mouse injury models [40,41]. The pipeline could also improve lncRNA coverage in poorly annotated animal models of cardiovascular disease. High turnover of lncRNA sequences during evolution means conserved lncRNAs are rare despite their high potential for function [15]. Expanding the lncRNA annotation of human and animal models is key to maximise discovery of such relationships. For example, lncRNA annotation in rat VSMCs stimulated with pro-inflammatory Angiotensin-II has extended coverage of Angiotensin-II-responsive lncRNAs in rats [42]. Using a matching approach in human VSMCs would allow comprehensive detection of rat-human conserved lncRNAs, which could be characterised in vivo in rats to provide relevant data for clinical relevance in human. For the same reason, it would be beneficial to match the lncRNAs highlighted in our study to orthologous lncRNAs in analogous animal models of VSMC proliferation, with our study providing a template methodology to achieve this.

Our use of an expression atlas to define cell-type enrichment in the VSMC for the expanded lncRNA repertoire aids predictions of lncRNA location and function in the vessel wall. The repertoire showed highest enrichment values within mesenchymal cells, including VSMCs. We identify 37 VSMC-enriched lncRNA which responded to either IL-1α/PDGF-BB or stiff-culturing (with 17 of these obtained through novel transcript discovery) and show their expression appears largely limited to VSMCs with a secondary tendency for expression in arterial adventitial fibroblasts. This could reflect the inherent similarity between VSMCs and fibroblasts or also be indicative of the fibroblast-like transcriptional profile that phenotypically modulated VSMCs have been observed to take on in vivo [25]. We show that a majority of these lncRNAs have minimal expression in other cell types involved in vessel wall remodelling, such as endothelial cells or leukocytes, which could be of value therapeutically. For example, targeting VSMC proliferation whilst preserving the endothelial barrier as a protective layer could be an effective strategy to improve the clinical outcome of late vein graft failure [43].

Similarly to cell-type specific lncRNAs, those with stimuli-specific expression are likely missing from reference annotation which cannot cover all biological conditions [13]. We see a particularly high proportion of newly assembled lncRNAs in induced clusters of genes associated with cytokine/chemokine response (46% newly assembled) and ribosomal/mitochondrial/Cajal activity suggestive of increased biosynthesis possibly related to proliferation (33% newly assembled). This could be explained by a low representation of these pathways in GENCODE. If so, many newly assembled lncRNAs may represent stimulus-specific lncRNAs, a trait that could in turn explain why newly assembled lncRNAs generally showed a high tendency to be differentially expressed. Further studies are required to study these stimuli-induced lncRNAs in non-VSMCs to determine if their expression is activated by the same stimulus on other cell types.

FANTOM and GeneHancer databases were used in this study to validate the structure of the identified lncRNAs and provide further characterisation in terms of expression or location relative to enhancer regions. Further analyses would benefit from using datasets matching to the VSMC type/stimuli in the datasets rather than these generic databases. For instance, no CAGE sites were found in FANTOM data for 26% of newly assembled lncRNAs. However, some of these missing TSSs could be identified using CAGEseq from VSMCs in the same context and these may represent particularly specifically expressed lncRNAs. Similarly, the identification of elncRNAs and their paired PCGs could be improved by studying chromatin marks in the same VSMC context as the RNAseq and by applying recently-developed, to link promoters and enhancers [44].

The tendencies identified amongst newly assembled genes for differential expression, cell-specificity and enhancer association are in broad agreement with other human lncRNA annotation efforts describing skin psoriasis [18] and cell state transitions [16,17]. We underscore the value of using such pipelines to highlight areas not yet covered by reference annotation, even in the extensively annotated human genome. RNAseq is an unbiased approach in comparison to microarray technology yet is under-utilised if relying solely on predefined reference annotation. An ever-increasing amount of high-quality sequencing data is available to profile lncRNA expression in a similar manner.

Reference annotation cannot capture the full transcriptional variety of all cellular states at all times. Therefore, focused annotation efforts such as in this study allow capturing missing details. We reinforce that these missing details provide important definition by identifying key components of the lncRNAs associated with VSMC pathology. We demonstrate an easily implemented approach to achieve this and provide a resource to help identify key candidate regulators of VSMC pathological states for investigation. We recommend similar strategies to comprehensively map lncRNA and maximise knowledge of their function in homeostasis and disease more broadly, as well as their potential for therapeutic manipulation.

## 4. Materials and Methods

See the online Appendix A for full methodology.

### 4.1. Transcriptome Assembly

Sequencing files were obtained from gene expression omnibus (GEO) using accession numbers GSE69637 and GSE100081. We received files for a third dataset of plaque VSMCs produced by Alloza et al. [22] by direct transfer from the authors. Data quality were checked via FastQC (version 0.11.9) [45]. Trimming of adaptor sequences was required for GSE69637 and done using TrimGalore (version 0.5.0) [46]. Custom transcriptomes consisting of GENCODE transcripts supplemented with newly assembled transcripts were generated as follows for each dataset. STAR (version 2.5.1b) [47] was used to map reads to the human genome (GRCh38) indexed with GENCODEv26 (sjdbOverhang 100). StringTie (version 1.3.1c) [48] was then used on these alignments to assemble transcripts (minimum length: 300 bp). The StringTie assemblies were merged (using StringTie—merge) with a filtered reference set for each sample to create an expanded transcriptome. Newly assembled transcripts and genes were provided numeric identifiers with the prefix “MSTRG.”. The filtered reference set was obtained by removing transcripts with low expression (<0.5 FPKM for spliced transcripts and < 1 FPKM for unspliced transcripts) and short transcripts (<300 bp). Transcript quantification was based on RSEM (version 1.3.0) [49] (bowtie2).

### 4.2. Pipeline for Annotation of LncRNA (PLAR) + Classification of Transcripts

RSEM [49] was used to quantify transcripts in each dataset using their corresponding expanded transcriptome. To annotate lncRNAs, we used the published pipeline PLAR [23,50]. For downstream analysis, we considered only expressed transcripts, defined as those with an average FPKM of at least 1 in an experimental condition. We also discarded minor isoforms for each newly assembled lncRNA gene, removing isoforms with an expression corresponding to less than 10% of the sum of all isoforms. Genes were classified based on PLAR transcript classification; genes producing any number of coding transcripts were labelled “coding”, remaining genes producing putative lncRNAs (see above) were labelled “putative lncRNAs” and remaining genes producing high confidence lncRNA were labelled as “lncRNAs”. Genes were considered robustly expressed if they had an average FPKM > 1 in 1 or more conditions in any dataset.

## Figures and Tables

**Figure 1 ijms-22-01484-f001:**
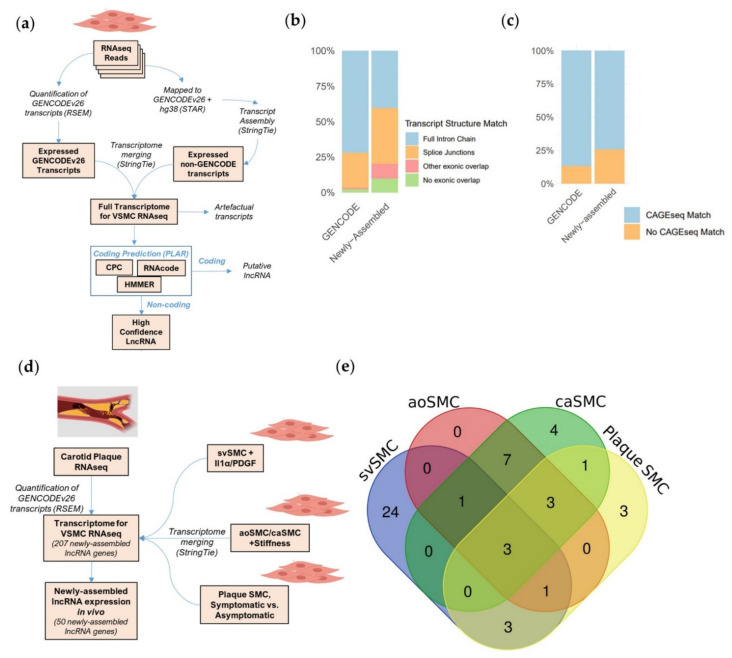
Identification of a high-confidence lncRNA repertoire for vascular smooth muscle cells (VSMCs) in physiological and pathological states. (**a**) Strategy to supplement GENCODE annotation with newly assembled transcripts and annotate lncRNAs. (**b**) Proportion of GENCODE and newly assembled lncRNA transcripts with structures matching FANTOM CAT annotation. (**c**) Proportion of GENCODE and newly assembled lncRNAs with a matching CAGE sequencing (CAGEseq) site in FANTOM CAT database. (**d**) Method to detect newly assembled lncRNAs in carotid plaque RNAseq. (**e**) Newly assembled lncRNAs derived from VSMC datasets detected in whole plaque tissue.

**Figure 2 ijms-22-01484-f002:**
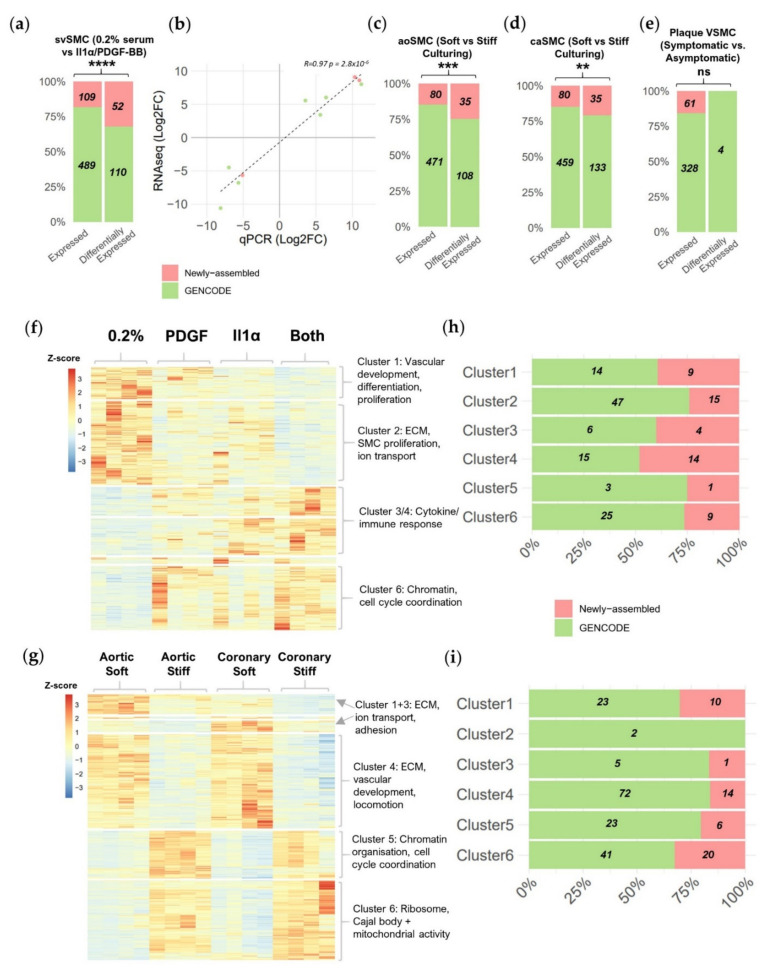
Newly assembled lncRNAs show a tendency to respond to pro-mitogenic stimuli. (**a**) Proportion of GENCODE and newly assembled lncRNAs expressed or differentially expressed in the saphenous vein VSMC (svSMC) dataset (Fisher’s exact test, background of all expressed genes). (**b**) Validation of expression dynamics in the svSMC RNAseq dataset by qRT-PCR for six lncRNAs (Spearman’s rank, *p* = 6.7 × 10^−4^). (**c**–**e**) Same as (**a**) but for remaining VSMC datasets. Expression heatmap of differentially expressed genes expression responding to (**f**) interleukin-1α (IL-1α)/platelet-derived growth factor-BB (PDGF-BB) or (**g**) stiff-culturing clustered hierarchically based on their variation across all samples with selected over-represented gene ontologies derived for each cluster. Proportion of GENCODE and newly assembled lncRNAs in each cluster of differentially expressed genes responding to (**h**) IL-1α/PDGF or (**i**) stiff-culturing. (**a**,**c**–**e**) *p* < 0.0001 ****, *p* < 0.001 ***, *p* < 0.01 **, ns not significant.

**Figure 3 ijms-22-01484-f003:**
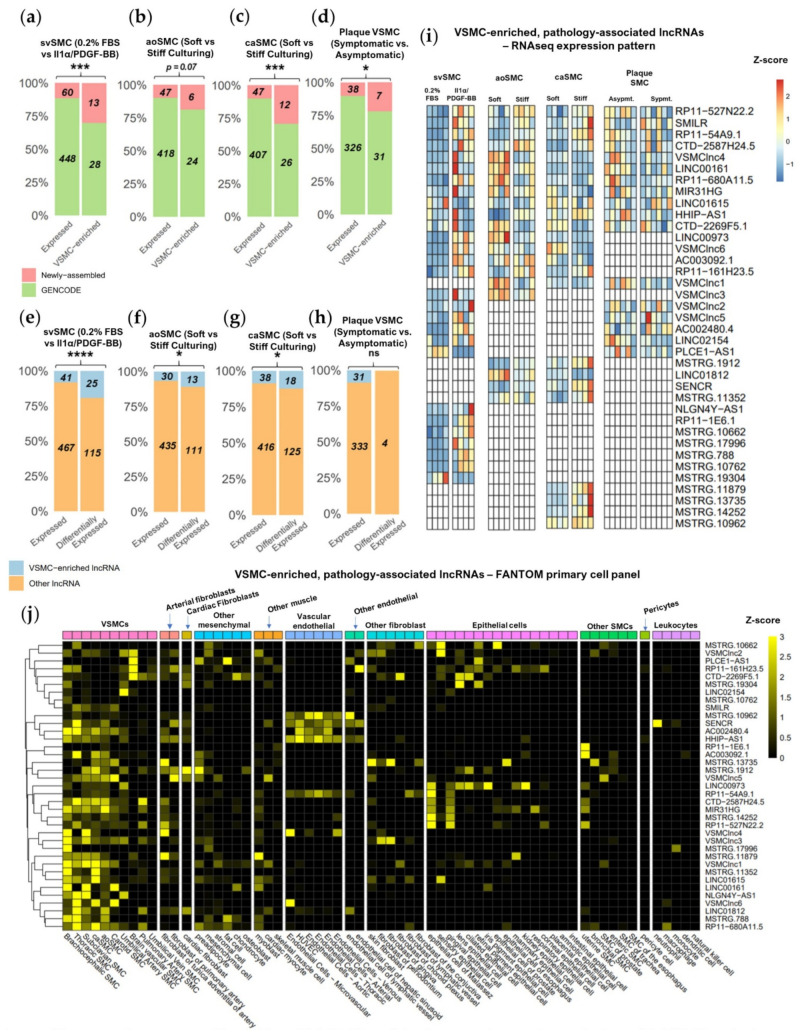
Increased coverage of lncRNAs with VSMC-enriched expression and association with VSMC pathology. (**a**–**d**) Proportion of newly assembled vs. GENCODE lncRNAs within CAGE-matched lncRNAs and CAGE-matched, VSMC-enriched lncRNAs. (**e**–**h**) Proportion of VSMC-enriched or non VSMC-enriched lncRNAs that are expressed or differentially expressed (Fisher’s exact test for (**a**–**h**), background of CAGE-matched lncRNAs). (**i**) Expression heatmap of all 37 VSMC-enriched and differentially expressed lncRNAs responding to IL-1α/PDGF-BB or stiff culturing as well as within plaque VSMCs (Z score generated for each VSMC type individually, white cells indicate no robust expression). (**j**) Expression heatmap of all 37 VSMC-enriched and differentially expressed lncRNAs across vascular cell types and other relevant cell types in the FANTOM CAGE expression atlas. (**a**–**h**) *p* < 0.0001 ****, *p* < 0.001 ***, *p* < 0.05 *, ns not significant.

**Figure 4 ijms-22-01484-f004:**
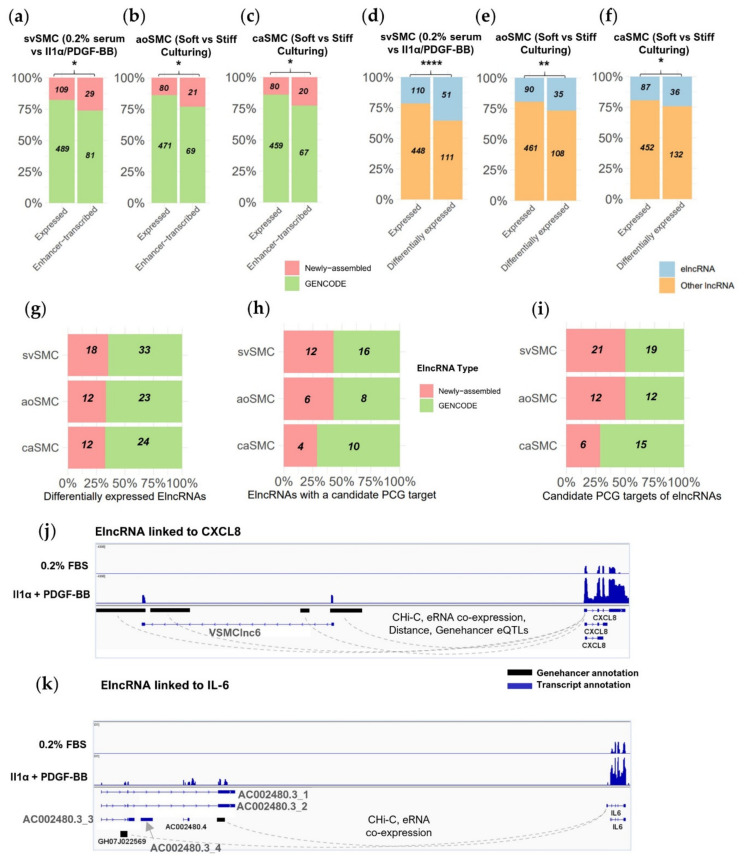
Increased coverage of elncRNAs and their association with VSMC pathology. (**a**–**c**) Proportion of newly assembled vs. GENCODE lncRNAs for expressed lncRNAs or elncRNAs in the three in vitro VSMC datasets (Fisher’s exact test, background of expressed lncRNAs). (**d**–**f**) Proportion of elncRNA vs. other lncRNAs amongst expressed or differentially expressed lncRNAs (Fisher’s exact test, background of expressed lncRNAs). (**g**) Proportions of newly assembled vs. GENCODE differentially expressed elncRNAs (**h**) Proportions of newly assembled vs. GENCODE elncRNAs with a candidate PCG target. (**i**) Proportion of candidate PCG targets identified for newly assembled and GENCODE elncRNA (any proximal to both types are considered as GENCODE). (**j**,**k**) Representative RNAseq coverage at two genomic regions with elncRNAs (AC002480.3 and VSMClnc6) linked to cytokine (*IL-6*) and chemokine (*CXCL8*) genes by GeneHancer interaction data. (**a**–**f**) *p* < 0.0001 ****, *p* < 0.01 **, *p* < 0.05 *.

**Table 1 ijms-22-01484-t001:** Differentially expressed elncRNAs with predicted protein-coding gene (PCG) targets and links to VSMCs, proliferation and/or migration.

ElncRNA Name ^1^	Interaction Evidence	PCG	VSMC Type + Stimulus	Stimulus Response	Max. FANTOM VSMC Enrichment	PCG VSMC Characterisation
*VSMClnc6* (svSMC + stiff-culturing)	CHi-C, eRNA co-expression, Distance, GeneHancer eQTLs	*CXCL8*	svSMC + IL-1α/PDGF-BB	Co-induced	LncRNA: 41-fold PCG: None	Activates VSMC proliferation + migration [30,31]
*AC002480.3*	CHi-C, eRNA co-expression	*IL-6*	svSMC + IL-1α/PDGF-BB	Co-induced	LncRNA: None (overlaps AC002480.4: 20-fold) PCG: 19-fold	Activates VSMC proliferation/migration + osteoblast phenotype [32,33,34]
*LINC00973*	FANTOM eQTL analysis	*DCBLD2*	All datasets	Co-induced with IL-1α/PDGF-BB Co-repressed with stiffness	LncRNA: 7-fold PCG: 5-fold	Regulates PDGFR + Il-8 expression, Marker of vascular remodelling [35]
*AC009229.5*	FANTOM eQTL analysis	*CYP1B*	aoSMC + stiffness	Co-repressed	LncRNA: None PCG: None	Mediates angiotensin II-induced VSMC proliferation + migration [36]
*MSTRG.10933* (svSMC)	Distance, GeneHancer eQTLs	*GLS*	svSMC + IL-1α/PDGF-BB	Co-repressed	LncRNA: None PCG: None	Required for TGFβ-induced myofibroblast differentiation + pro-fibrotic marker expression [37]
*NR2F2-AS1*	GeneHancer eQTLs, CHi-C	*NR2F2 (COUP-TFII)*	svSMC + IL-1α/PDGF-BB	Co-repressed	LncRNA: None PCG: None	Ablation leads to osteoblast-like phenotype in mesenchymal precursors [38]

^1^ Newly assembled lncRNAs given name (see methods) along with their respective RNAseq dataset in brackets.

## Data Availability

The authors declare all supporting data are available within the article, the online supplementary files, and at https://bakergroup.shinyapps.io/VSMClncRNAannotation/.

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
