# Peer review of "Novel Transcript Discovery Expands the Repertoire of Pathologically-Associated, Long Non-Coding RNAs in Vascular Smooth Muscle Cells"

_ijms, 2021, doi:10.3390/ijms22031484_

Round 1
Reviewer 1 Report
In their manuscript Bennett et al. apply an in silico strategy to identify lncRNAs at a genome-wide and tissue-specific (vascular smooth muscle cells) scale. The used strategy itself is sound and might prove useful to test the biological role of the identified lncRNA candidates in future studies. In respect to the poor annotation of lncRNA genes it is important to do tissue specific profiling and in depth analysis of RNAseq data sets. The manuscript itself is exceptionally well written und understandable. I recommend this manuscript for publication in IJMS.
Author Response
Many thanks for the kind words and feedback
Reviewer 2 Report
Novel transcript discovery expands the repertoire of pathologically-associated, long non-coding RNAs in vascular smooth muscle cells.
I must congratulate authors for great effort and novel discovery. Yet requiring full validation and in-depth investigation, data provide wide field for further research and discovery. Definitely, main strength of this publication is computation analysis which is well executed by the team.
The VSMC as stromal cells might be involved in many pathological processes in the vessel wall. Suggest to add very nice recently published review by Shanashan et all (CM Shanahan et al Frontiers in Immunology 11, 3053. Role of Vascular Smooth Muscle Cell Plasticity and Interactions in Vessel Wall Inflammation)
The role of miRNA more investigated in murine model, however human VSMC need additional test and analysis. It would be great addition to manuscript if this difference clearly stated in the manuscript (CC Woo…International journal of molecular sciences 21 (1), 11)The interaction between 30b-5p miRNA and MBNL1 mRNA is involved in vascular smooth muscle cell differentiation in patients with coronary atherosclerosis.
Although lncRNA is important, other epigenetic mechanism should not be downplaying and should be discussed/mention in this context. (R Gurung.et al. International Journal of Molecular Sciences 21 (17), 6334. Genetic and Epigenetic Mechanisms Underlying Vascular Smooth Muscle Cell Phenotypic Modulation in Abdominal Aortic Aneurysm)
Author Response
Many thanks for the kind words and feedback. The reviewer raises some important points which will help to provide further context on the possible role of these lncRNAs in possible future characterisations. We have accordingly updated the introduction and discussion section to cover the following points.
As suggested, we have added the recent review by Shanahan et al. in the introduction (lines 39+47 in the revised manuscript) when describing VSMC plasticity in biological and pathological conditions and in the discussion (lines 428-431 in the revised manuscript) when describing the pro-inflammatory transcriptional activity profiled in our datasets of our revised manuscript.
We agree that data obtained in a specific animal model needs to be investigated and confirmed across species. Contrary to miRNA, lncRNAs are not more investigated in murine model. We believed the lncRNAs identified in human VSMCs that are conserved in a different species will need to be functionally characterised both in human and in a relevant species model. Conversely, transcriptomics analysis of VSMCs from other species might provide candidates with relevance to human after further investigation. Our revised manuscript discussion has updated text reflecting these two points in lines 451-465:
“The pipeline could also improve lncRNA coverage in poorly annotated animal models of cardiovascular disease. High turnover of lncRNA sequences during evolution means conserved lncRNAs are rare despite their high potential for function[14]. Expanding the lncRNA annotation of human and animal models is key to maximise discovery of such relationships. For example, lncRNA annotation in rat VSMCs stimulated with pro-inflammatory Angiotensin-II has extended coverage of Angiotensin-II-responsive lncRNAs in rats[42]. Using a matching approach in human VSMCs would allow comprehensive detection of rat-human conserved lncRNAs which could be characterised in vivo in rats to provide relevant data for clinical relevance in human. For the same reason, it would be beneficial to match the lncRNAs highlighted in our study to orthologous lncRNAs in analogous animal models of VSMC proliferation, with our study providing a template methodology to achieve this.”
We have also updated the manuscript to put lncRNAs in context with other important epigenetic mechanisms for VSMC modulation, including the suggested reference, within the introduction at lines 45-47:
“The molecular mechanisms controlling the wide variety of phenotypic changes involved in such diseases are controlled by both coding and non-coding genes at both the genetic and epigenetic level[3,4]. However they are not yet understood sufficiently to effectively target them therapeutically. “
And also in the discussion at lines 445-447:
“ElncRNAs are only one of many types of components involved in changing the chromatin accessibility at sites associated with VSMC pathology[4]. Therefore, any further characterisation of the elncRNAs highlighted here must put their potential mechanism in context with any recruitment of transcription factors or histone-modifying enzymes.”